# SARS-CoV-2 Vaccination and Clinical Presentation of COVID-19 in Patients Hospitalized during the Delta- and Omicron-Predominant Periods

**DOI:** 10.3390/jcm12030961

**Published:** 2023-01-26

**Authors:** Daša Stupica, Stefan Collinet-Adler, Nataša Kejžar, Mario Poljak, Tina Štamol

**Affiliations:** 1Department of Infectious Diseases, University Medical Center Ljubljana, Japljeva 2, 1525 Ljubljana, Slovenia; 2Department of Infectious Diseases, Faculty of Medicine, University of Ljubljana, Japljeva 2, 1000 Ljubljana, Slovenia; 3Department of Infectious Diseases, Methodist Hospital, Park Nicollet/Health Partners, Minneapolis, MN 55426, USA; 4Institute for Biostatistics and Medical Informatics, Faculty of Medicine, University of Ljubljana, Vrazov Trg 2, 1000 Ljubljana, Slovenia; 5Institute of Microbiology and Immunology, Faculty of Medicine, University of Ljubljana, Zaloška 4, 1000 Ljubljana, Slovenia

**Keywords:** COVID-19 vaccine, vaccine breakthrough, SARS-CoV-2 antibodies

## Abstract

Evidence suggests that monovalent vaccine formulations are less effective against the Omicron SARS-CoV-2 than against previous variants. In this retrospective cohort study of hospitalized adults with PCR-confirmed COVID-19 during the Delta (October–November 2021) and Omicron (January–April 2022) variant predominant periods in Slovenia, we assessed the association between primary vaccination against SARS-CoV-2 and progression to critically severe disease (mechanical ventilation or death). Compared with the 529 patients hospitalized for acute COVID-19 during the Delta period (median age 65 years; 58.4% men), the 407 patients hospitalized during the Omicron period (median age 75 years; 50.6% men) were older, more often resided in long-term care facilities, and had higher Charlson comorbidity index scores. After adjusting for age, sex, the Charlson comorbidity index, the presence of immunocompromising conditions, and vaccination status, the patients admitted during the Omicron period had comparable odds of progressing to critically severe disease to those admitted during the Delta period. The 334/936 (35.7%) patients completing at least primary vaccination had lower odds of progression to critically severe disease and shorter hospital stay than unvaccinated patients; however, the protective effect of vaccination was less pronounced during the Omicron than during the Delta period. Although the Omicron variant appeared to better evade immunity induced by monovalent vaccines than the Delta variant, vaccination against SARS-CoV-2 remained an effective intervention to decrease morbidity and mortality in COVID-19 patients infected with the Omicron variant.

## 1. Introduction

Safe and effective vaccines against severe acute respiratory syndrome coronavirus 2 (SARS-CoV-2) have provided significant protection from the coronavirus disease 2019 (COVID-19) [1,2,3,4]. Nonetheless, the emergence of successive variants and variable time intervals since vaccination and SARS-CoV-2 infection have been accompanied by breakthrough COVID-19 severe enough to require hospitalization [5,6,7]. The successive appearance of viral mutations, waxing and waning disease- and vaccine-derived immunity, improved therapeutics, and variably effective and sustainable public health measures make assessments of differential vaccine efficacy against each SARS-CoV-2 variant particularly challenging [7,8]. In November 2021, a fifth SARS-CoV-2 variant of concern was reported [9], designated as Omicron [10]. Evidence suggests that patients infected with the Omicron variant present with reduced severity of disease [11,12] and lower mortality rates after controlling for patient demographics, previous infection, and vaccination status [13,14] compared with those infected with the Delta variant. Additionally, monovalent vaccines appear less effective against Omicron than against previous variants [15,16]. Firm conclusions based on these studies are limited due to the potential misclassification of cases, microbiologic ascertainment biases, and missing information on vaccination status and disease outcomes. Specifically, information on the effect of primary vaccination on disease outcomes in hospitalized patients with PCR-confirmed COVID-19 based on individual-level data is lacking.

In this retrospective, observational cohort study of patients hospitalized for PCR-confirmed COVID-19 during the Delta- and Omicron-predominant periods, we investigated the clinical course of disease and protection against progression to critically severe disease (mechanical ventilation or death) conferred by primary vaccination with two or three doses of monovalent SARS-CoV-2 vaccines. We also evaluated serum antibody titers to SARS-CoV-2 in a subpopulation of fully vaccinated patients hospitalized with vaccine breakthrough SARS-CoV-2 infection during both periods.

## 2. Materials and Methods

### 2.1. Design and Setting

Adult patients (≥18 years) hospitalized with COVID-19 at the main university medical center in Ljubljana, Slovenia, between 1 October and 23 November 2021 (Delta period) and between 15 January and 8 April 2022 (Omicron period) were assessed for enrollment on 10 October 2022. STROBE guidelines for reporting were followed. We compared COVID-19 severity and the length of hospitalization between patients hospitalized during the Delta- and Omicron-predominant periods. We also assessed serologic responses to SARS-CoV-2 in a subgroup of vaccinated patients hospitalized with breakthrough COVID-19 during both periods.

### 2.2. Participants

Hospitalized patients were screened for potential eligibility through the hospital’s electronic medical record system. Patients who had a positive SARS-CoV-2 PCR test within 10 days before admission or during the hospital stay were eligible. Those hospitalized with a clinical presentation consistent with acute COVID-19 were defined as COVID-19 cases [17]. Patients hospitalized for medical conditions unrelated to COVID-19 were excluded from the analysis. Patients who were partially vaccinated against SARS-CoV-2 were excluded from the analysis of vaccine breakthrough cases.

### 2.3. Data Collection

Demographic, clinical, and laboratory data were collected through hospital medical record review and by analyzing remnant clinical specimens after standard-of-care testing. The national electronic central vaccine registry was used to verify dates of SARS-CoV-2 vaccination, vaccine type, and lot numbers.

### 2.4. Laboratory Analysis

The presence of SARS-CoV-2 RNA in upper respiratory specimens was determined using one of two fully automated rtRT-PCR analyzers: Cobas 6800 (Roche Diagnostics, Basel, Switzerland) or Alinity m (Abbott, Chicago, IL, USA), as described previously [18]. The presence of total antibodies to the SARS-CoV-2 spike (S) protein receptor-binding domain was determined using automated electrochemiluminescence immunoassay Elecsys Anti-SARS-CoV-2 S (Roche Diagnostics, Basel, Switzerland), as described previously [19]. Tests were performed and results were interpreted according to the manufacturer’s instructions.

### 2.5. Classification of Vaccination Status and COVID-19 Severity

Patients were considered fully vaccinated if they received the second dose of BNT162b2, mRNA-1273, or ChAdOx-1S or the first dose of Ad.26.COV2.S at least 14 days before symptom onset [20]. If patients received an additional dose of BNT162b2 or mRNA-1273, this was considered a booster dose. Participants were considered unvaccinated if they had received no vaccine doses before symptom onset or partially vaccinated if they had received one dose of BNT162b2 or mRNA-1273 or two doses of BNT162b2 or mRNA-1273 or one dose of Ad.26.COV2.S within 14 days before symptom onset. We evaluated vaccination status both as a binary variable (vaccinated or not) and as a continuous variable (time in weeks from the last vaccine dose). Critically severe disease was defined as progression to invasive mechanical ventilation or death, corresponding to levels 7, 8, 9, and 10 of the World Health Organization’s COVID-19 Clinical Progression Scale [21].

### 2.6. Statistical Analysis

Categorical data were summarized as frequencies (%) and numerical data as medians (interquartile range, IQR). Differences between the groups (according to vaccination) were tested using the Mann–Whitney test (numerical) or Fisher’s exact test (categorical variables). The association between progression to critically severe disease and a predetermined set of covariates including full vaccination status was estimated using multiple logistic regression. The results are presented as odds ratios (ORs) with 95% confidence intervals (CIs). The association between vaccination and the probability of discharge was calculated by a cause-specific Cox model and supplemented by Fine–Gray analysis [22]. To further account for potential confounding, a sensitivity analysis was performed using a propensity score analysis that matched vaccinated and unvaccinated patients in terms of age, sex, the Charlson comorbidity index, the presence of an immunocompromising condition, and the study period (Delta vs. Omicron). R statistical language (version 4.1.1) was used for the analyses, R package cmprsk was used for the Fine–Gray analysis, mstate for the cause-specific Cox time-to-event model, and MatchIt for the propensity score matched dataset.

## 3. Results

### 3.1. Characteristics of Hospitalized Patients with COVID-19

Patients enrolled between 1 October 2021 and 23 November 2021 and between 15 January 2022 and 8 April 2022 were admitted to the hospital during the Delta and Omicron variant predominant periods in Slovenia, respectively [23,24]. Among the 566 patients hospitalized with PCR-confirmed SARS-CoV-2 infection during the Delta period (median age 66 years (IQR 53.3–79); 331 (58.5%) men), 529 patients (93.5%) were hospitalized due to acute COVID-19 and 37 patients (6.5%) for indications unrelated to COVID-19 (Figure 1).

Among the 795 patients admitted with SARS-CoV-2 infection during the Omicron period (median age 72 years (IQR 60–82); 393 (49.4%) men), the proportion of those hospitalized due to COVID-19 was significantly lower (407/795 (51.2%)) than that during the Delta period (*p* < 0.001).

As shown in Table 1, those hospitalized for acute COVID-19 during the Omicron period were older, more often resided in long-term care facilities, and had higher Charlson comorbidity index scores than patients hospitalized for acute COVID-19 during the Delta period. The proportion of patients with hypoxemia at admission was lower during the Omicron than during the Delta period (328/407 (80.6%) vs. 485/529 (91.7%); *p* < 0.001). Most admitted patients (≥84%) were treated with corticosteroids during both study periods, but less frequently during the Omicron period. Remdesivir was prescribed more often during the Omicron period (Table 1). The median time from admission to progression to critically severe disease was 5.4 days (IQR 3.2–9.0) during the Omicron period and 4 days (IQR 2.0–8.1) during the Delta period (*p* = 0.025). At the time of evaluation (10 October 2022), all except one patient were discharged or deceased. This patient had been hospitalized for 139 days, survived mechanical ventilation, and was considered critically ill in the analysis.

### 3.2. Association of SARS-CoV-2 Variant and Vaccination Status with Progression to Critically Severe Disease

Patients admitted due to acute COVID-19 during the Omicron-predominant period had comparable odds of progressing to critically severe disease (invasive mechanical ventilation or death) to patients admitted during the Delta-predominant period (Table 2). The model was adjusted for age, sex, the Charlson comorbidity index, the presence of immunocompromising conditions, and vaccination status. Fully vaccinated patients were older (median 76 years (IQR 67–84) vs. 65 years (IQR 53–78); *p* < 0.001), had higher Charlson comorbidity indices (median 4 (IQR 3–6) vs. 3 (1–4); *p* < 0.001), and more often had an accompanying immunocompromising condition (57/334 (17.1%) vs. 30/595 (5.0); *p* < 0.001) but did not significantly differ in regard to sex (male 197/334 (59.0%) vs. 314/595 (52.8%); *p* = 0.074) or the frequency of hypoxemia on admission (280/334 (83.8%) vs. 527/595 (88.6%); *p* = 0.044), compared with unvaccinated patients (Appendix A). In order to simplify the model and avoid overfitting, diabetes and certain pulmonary and cardiovascular diseases already included in the Charlson comorbidity index were not separately included. Increasing age and higher Charlson comorbidity index were associated with higher odds of progressing to critically severe disease (Table 2).

As the effect of vaccination could be different for different SARS-CoV-2 variants, we also included the interaction between these two variables in the model. Those completing at least primary vaccination had lower odds of progressing to critically severe disease than unvaccinated patients, regardless of the SARS-CoV-2 variant (Table 2). However, the protective effect of vaccination against progressing to critically severe disease was less pronounced in our dataset during the Omicron than during the Delta period (Figure 2).

To analyze the duration of hospital stay, a cause-specific Cox model with the competing event of death was fitted to the data (Table 3). After adjusting for selected variables with potential influence on disease course (SARS-CoV-2 variant, sex, age, the Charlson comorbidity index, and the presence of immunocompromising condition), vaccination was significantly associated with discharge (HR 1.39 [95% CI 1.13–1.72]; *p* = 0.002). Since death happens earlier than discharge in this population, HR > 1 correlates with a shorter hospital stay. The patients who were immunocompromised stayed significantly longer in the hospital.

In a supplementary analysis, propensity score matching was used to estimate the average marginal effect of vaccination on progression to critically severe disease, accounting for confounding by age, sex, the Charlson comorbidity index, the presence of immunocompromising conditions, study period (Omicron vs. Delta), and the interaction between vaccination status and virus variant. The odds of progression to critically severe disease were smaller in vaccinated than in unvaccinated patients (OR 0.52 [95% CI 0.30–0.90]; *p* < 0.019).

### 3.3. COVID-19 in Vaccine Breakthrough Cases

Among the 529 patients hospitalized with acute COVID-19 during the Delta-predominant period, 175 patients (33.1%) were fully vaccinated. The median time from the last primary vaccine dose to hospital admission was 27 weeks (IQR 21.3–33.5). The proportion of fully vaccinated patients hospitalized with acute COVID-19 during the Omicron-predominant period was non-significantly higher (159/407 (39.1%) vs. 175/529 (33.1%); *p* = 0.063). Only eight patients, all infected during the Omicron period, self-reported previous COVID-19 infection. The time from the last primary vaccine dose to hospital admission during the Omicron period was longer than during the Delta period (median 37.9 weeks (IQR 27–47.1) vs. median 27 weeks (IQR 21.3–33.5); *p* < 0.001) (Table 4). During the Omicron period, patients more often reported having received a booster dose than during the Delta period. For this reason, the difference in the time since the last vaccination dose between the two study periods was reversed when the timing of booster doses was considered (Table 4).

We performed an additional regression model to explore a possible association between the time elapsed since receipt of the last vaccine dose, regardless of whether this was a primary or booster vaccination, and progression to critically severe disease (adjusting for age, sex, the Charlson comorbidity index, and the study period). No significant association was found; however, caution should be exercised with this analysis since the model was not well calibrated (Appendix A).

### 3.4. SARS-CoV-2 Antibody Testing

The results of anti-SARS-CoV-2 antibody testing were available at admission to the hospital in 141/175 (80.6%) and 109/159 (68.6%) of the fully vaccinated patients hospitalized for breakthrough COVID-19 during the Delta and Omicron periods, respectively. Antibodies to the SARS-CoV-2 S-RBD were detected in 134/141 (95.0%) and 102/109 (93.6%) of the tested patients and reached values above 2500 BAU/mL in 86/141 (61.0%) and 60/109 (55.0%) of the patients during the Delta and Omicron periods, respectively. In comparison with patients with low (≤2500 BAU/mL) anti-SARS-CoV-2 antibody titers, those with high (>2500 BAU/mL) antibody titers progressed to critically severe disease comparably often (22/146 (15.1%) vs. 25/104 (24.0%); *p* = 0.100) but had a lower mortality rate (14/146 (9.6%) vs. 20/104 (19.2%); *p* = 0.039). Those with high titers of anti-SARS-CoV-2 antibodies had lower Charlson comorbidity indices (median 4 (IQR 3–5.8) vs. 5 (IQR 3–6.2); *p* < 0.017). Age was not a significant factor differentiating between the high-titer group (median 75 years (IQR 65.2–83)) and the low-titer group (median 76 years (IQR 65–83)) (*p* = 0.806).

The median values of anti-S-RBD antibody titers during the Delta period were higher than those during the Omicron period, but the difference was not statistically significant (6816 BAU/mL (IQR 369–43,650) vs. 2848 BAU/mL (IQR 415–18,580); *p* = 0.204). In the additional exploratory logistic regression, the values of anti-S-RBD antibody titers were not significantly associated with progression to critically severe disease when adjusting the analysis for age, sex, the Charlson comorbidity index, the presence of immunocompromising conditions, and the viral variant (Appendix A).

## 4. Discussion

Of 936 adult patients hospitalized due to COVID-19, those admitted from October to November 2021, when the Delta variant predominated, had comparable odds of progression to critically severe disease as those hospitalized from January to April 2022, when the Omicron variant was predominant, after adjusting for age, sex, the Charlson comorbidity index, and the presence of immunocompromising conditions. During both study periods, vaccinated patients had a lower risk for progression to critically severe disease and a shorter length of hospital stay than unvaccinated patients, but the protective effect of vaccination was less pronounced during the Omicron than during the Delta period.

Several investigators showed lower hospital admission rates and less severe disease or lower mortality for the Omicron period, compared with the Delta period [11,13,14,15,16,25]. Studies describing disease course and outcome specifically among hospitalized patients also showed lower severity and/or mortality during the Omicron-predominant period compared with the Delta-predominant period [12,14,15,26]. Another report was not able to conclude about the comparative risk of severe outcomes in hospitalized patients relative to the COVID-19 period, possibly due to insufficient power [11].

We found that the patients admitted during the Omicron period had insignificantly lower odds of progression to critically severe disease than those admitted during the Delta period when adjusting for age, sex, the Charlson comorbidity index, and the presence of immunocompromising conditions. The Charlson comorbidity index is a score of 19 comorbidities weighted according to severity [27] and was used in our study as a measure of the severity of comorbid conditions with the intention of providing a more detailed description of the study population. Our results are in line with those reported by Fall et al., who found that the need for supplementary oxygen or admission to ICU in Omicron-infected patients (N = 1119) hospitalized for COVID-19 was not statistically significantly different from Delta-infected patients (N = 908) [15]. Bouzid et al. found that Omicron infection (N = 244) was associated with less risk for ICU-level care, mechanical ventilation, and mortality than Delta infection (N = 456) in patients seen in the emergency department for COVID-19 [26]. Many potential patients in this last study were excluded because they were discharged from the emergency department, possibly overestimating the risk of severe outcomes with Omicron.

In our study, the proportion of patients with hypoxemia on admission was high during both study periods (>80%) but was lower during the Omicron period. The statistically significant decrease in the need for any steroid therapy and high-dose methylprednisolone therapy during the Omicron-predominant period reflects this. There was also a non-significant trend towards less use of tocilizumab during the Omicron-predominant period. The decrease in hypoxemia seen during the Omicron period may be due to phenotypic changes in the viral tropism of Omicron with less efficient replication in the lungs [28] and/or suggest that in older people with COVID-19, symptoms other than hypoxemia may lead to hospital admission. Since autopsies were performed only exceptionally, we do not have detailed data for patterns of deaths beyond detecting a cardiorespiratory failure in the diseased.

In both population-based analyses and reports on hospitalized patients, Omicron-infected patients tended to be younger than those infected during the Delta period [11,14,15,25]. We found the opposite in our cohort. This discrepancy between studies may be partially associated with contrasting socio-demographics and healthcare delivery models in different countries, but it may also reflect dissimilar study designs. In our study, only SARS-CoV-2-positive patients hospitalized due to COVID-19 were analyzed. Those hospitalized for other reasons and found to have incidental SARS-CoV-2 infection were excluded. The higher median age of Omicron-infected patients in our study compared with Delta-infected patients might, in part, be due to the relatively better protective effectiveness of vaccines against hospitalization in younger and immunocompetent populations [29,30]. It is also possible that the decreased general severity of Omicron was such that pre-existing fragility, either through age and/or comorbidities, was proportionally more often necessary in this period to trigger illness severe enough to lead to hospitalization compared with the Delta-predominant period.

We found that the indicators of medical fragility were more frequently present in vaccinated hospitalized patients with COVID-19 than in unvaccinated patients. The promotion of vaccination against SARS-CoV-2 in patients with known risk factors for severe disease such as advanced age and comorbidities including immunosuppression is a high priority worldwide [31]. These individuals were also more likely to be vaccinated against SARS-CoV-2 in Slovenia [32]. In our study, the vaccinated patients hospitalized due to COVID-19 were older and more medically fragile than unvaccinated patients. These results are consistent with other studies of hospitalized COVID-19 cases [7,33] and may reflect the prioritization of vaccination in these groups. The study finding that a relatively high and growing proportion of the patients hospitalized due to COVID-19 were vaccinated against SARS-CoV-2 may simply reflect increasing vaccination coverage in the general population and should not be automatically interpreted as an indicator that vaccines are no longer effective. The lower proportion of vaccinated patients among those hospitalized due to COVID-19 compared with the vaccination coverage in the general population at the beginning of both study periods, particularly in older age groups, suggests that vaccination still has a protective value against hospitalization due to COVID-19. In the general Slovenian population, 79.9% of persons 65 years or older had received at least primary vaccination by 1 October 2021, and 87.1% by 15 January 2022, respectively. A further 60.8% of persons 65 years or older received a booster by 15 January 2022 [32].

COVID-19 vaccinations were shown to effectively prevent hospitalizations, serious illness, and death during both the Delta and Omicron periods [33,34]. Assessing vaccine effectiveness in preventing hospitalization due to COVID-19 is not within the scope of this study. Our results suggest that in the population of patients who were hospitalized due to COVID-19, primary vaccination with or without a booster provided protection against severe disease outcomes (progression to mechanical ventilation or death). The need for high-dose methylprednisolone and tocilizumab use was significantly lower in vaccinated individuals than in unvaccinated patients (Appendix A), again pointing to vaccine efficacy. Although the difference in the probability of progressing to critically severe disease between unvaccinated and vaccinated patients was smaller during the Omicron period than during the Delta period, vaccination remained protective during both study periods. Our results are in line with previous studies, which showed that vaccines attenuated disease severity in non-hospitalized [35] and hospitalized breakthrough infections [7] during the pre-Delta and Delta periods in the US. Vaccination was also protective against the progression to severe disease in patients hospitalized for COVID-19 symptoms in South Africa during the Delta and Omicron periods, but the difference in the magnitude of protection between the two periods was not evaluated [14]. Depending on variant characteristics, the use of bivalent vaccine formulations may further increase the value of COVID-19 vaccination [36].

Conversely, in another study using representative data from 11,127 laboratory-confirmed COVID-19-associated hospitalizations occurring from January 2021 to April 2022 (pre-Delta, Delta, and Omicron periods), no protective effect of vaccination against progression to severe disease (ICU admission or in-hospital death) was found. The results of the sub-analysis using the propensity score-matched cohort of 2000 vaccinated and 2000 unvaccinated patients were similar; vaccination was not significantly associated with a reduced risk of severe disease [33]. The authors reasoned that unidentified confounders may have led to these surprising results.

The duration of illness for non-hospitalized patients [35] and the duration of hospital stay for hospitalized patients [7,33] were reported to be shorter among patients who had breakthrough infection than unvaccinated patients, but these analyses were not performed to compare selected time periods with a single SARS-CoV-2 variant predominance. Our results suggest that among the different characteristics recognized to be associated with COVID-19 outcome, vaccination was the strongest predictor of shorter hospital stay regardless of the study period (Omicron vs. Delta); however, the beneficial effect of vaccination on the duration of hospital stay was less pronounced during the Omicron than during the Delta period.

In our study, patients with higher Charlson comorbidity indices more often had low anti-S-RBD antibody titers, and those with low titers had higher mortality rates than those with high titers, with the cutoff set at 2500 BAU/mL. However, we could not confirm that the magnitude of anti-S-RBD antibody titers was associated with progression to critically severe disease after adjustment for age, sex, comorbidities, immunocompromise, and SARS-CoV-2 variant. Antibody responses to the SARS-CoV-2 vaccine are variable and can be influenced by a number of factors, including age, comorbidities, immune function, number of vaccines, and the time since previous vaccination or SARS-CoV-2 infection [37,38,39]. Our data do not shed light on a specific protective level of antibody response against severe COVID-19 disease. It is possible that the use of other cutoff values to distinguish between low- and high-antibody anti-S-RBD titers may have led to a different conclusion.

The strengths of this study were that in all cases, SARS-CoV-2 infection was confirmed with PCR and that reliable information on vaccination status, baseline clinical characteristics, and outcome status was available for all individual patients, enabling the adequate classification of cases with regard to the reason for hospital admission and vaccination status and better control of confounding and effect modification.

This study has several limitations. First, due to the retrospective observational nature of the study, potential unmeasured confounding may have occurred, but we controlled for several relevant confounders, such as sex, age, and comorbidity status. Second, the study was limited to patients admitted to a university hospital, and we cannot conclude about the protective efficacy of vaccination against hospitalization in the general population. Third, we cannot exclude the possibility of admission bias in regard to disease severity and vaccine status, but this seems less likely since the proportion of patients admitted with hypoxemia was fairly comparable between the fully vaccinated and unvaccinated groups. Fourth, this was a single institution study in a relatively small region in central Europe with little demographic heterogeneity, thus limiting the generalizability of results. Fifth, we did not perform individual SARS-CoV-2 genome sequencing, so we cannot confirm that all the cases during each study period were due to the spread of the predominant SARS-CoV-2 variant, thereby potentially limiting the validity of some of our comparisons. We believe that this shortcoming likely had a very limited impact on our results since, as part of nationally coordinated sequencing efforts, more than 80% of randomly selected newly detected SARS-CoV-2 are routinely sequenced in Slovenia in any given week, and >95% of the cases were due to the predominant SARS-CoV-2 variant during the periods studied [23,24].

## 5. Conclusions

In this retrospective, observational cohort study of adult patients hospitalized due to PCR-confirmed COVID-19, the patients hospitalized during the Omicron period were older and had more chronic illnesses than those during the Delta period; nevertheless, the probability of progressing to critically severe disease was insignificantly lower during the Omicron period when adjusting the analysis for these confounders. During both study periods, vaccination against SARS-CoV-2 provided the most effective protection against adverse clinical outcomes in patients admitted to the hospital for COVID-19. This remains a relevant finding in the face of ongoing vaccine hesitancy and adds to the well-documented protective efficacy of COVID-19 vaccines against severe COVID-19-associated outcomes.

## Figures and Tables

**Figure 1 jcm-12-00961-f001:**
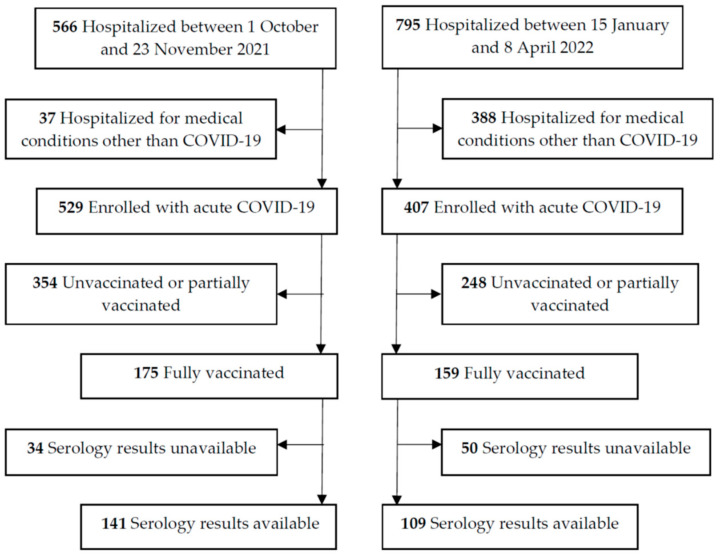
Flow diagram.

**Figure 2 jcm-12-00961-f002:**
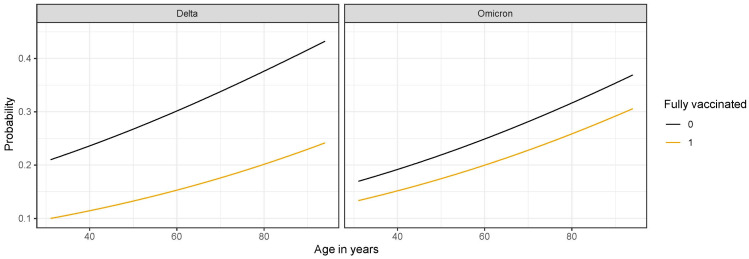
Modeled probability of progression to critically severe disease during the Delta and Omicron periods by vaccination status (vaccinated vs. unvaccinated) and age in years, as assessed from logistic regression model for two male patients with COVID-19, Charlson comorbidity index 4, and without immunocompromising conditions.

**Table 1 jcm-12-00961-t001:** Characteristics of patients hospitalized for COVID-19 by study period.

Characteristic	Delta Period ^1^	Omicron Period ^1^	*p*-Value ^2^
	*n* = 529/566 (93.5%) ^3^	*n* = 407/795 (51.9%) ^3^	
Age	65 (53–78)	75 (64–84)	**<0.001**
Men	309 (58.4)	206 (50.6)	0.02
Charlson comorbidity index	3 (1–4)	4 (3–6)	**<0.001**
Chronic illnesses ^4^			
Cardiovascular disease	298 (56.3)	281 (69.0)	**<0.001**
Pulmonary disease	67 (12.7)	70 (17.2)	0.062
Asthma	37 (7.0)	19 (4.7)	0.164
COPD	30 (5.7)	38 (9.3)	0.041
Diabetes type II	113 (21.4)	102 (25.1)	0.184
Obesity (body mass index ≥ 30)	183 (34.6)	118 (29.0)	0.078
Immunocompromising condition ^5^	52 (9.8)	36 (8.8)	0.652
One or more comorbidities	396 (74.9)	301 (74.0)	0.763
One comorbidity	82 (15.5)	58 (14.3)	0.644
Resident of long-term care facility	18 (3.4)	44 (10.8)	**<0.001**
Fully vaccinated ^6^	175 (33.1)	159 (39.1)	0.063
Received booster dose	1 (0.2)	55 (13.5)	**0.001**
Self-reported previous infection	0	8 (2.0)	**0.009**
Primary vaccine received			
BNT162b2	122 (69.7)	130 (81.8)	**0.009**
ChAdOx-1S	26 (14.9)	13 (7.8)	—
Ad.26.COV2.S	18 (10.3)	6 (3.6)	—
mRNA-1273	9 (5.1)	12 (7.2)	—
Therapy			
Remdesivir	55 (10.4)	68 (16.7)	**0.006**
Any corticosteroids	493 (93.2)	342 (84.0)	**<0.001**
Dexamethasone ^8^	398 (75.2)	272 (66.8)	**0.005**
Methylprednisolone ^8^	202 (38.2)	116 (28.5)	**0.002**
Tocilizumab	14 (2.6)	2 (0.5)	0.011
Monoclonal antibodies	14 (2.6)	18 (4.4)	0.15
Hypoxemic at admission	485 (91.7)	328 (80.6)	**<0.001**
Critically severe disease (WHO score 7–10) ^7^	125 (23.6)	107 (26.3)	0.36
Fully vaccinated	33 (26.4)	42 (39.3)	0.048
Unvaccinated	92 (73.6)	62 (57.9)	—
Death (WHO score 10) ^7^	72 (13.6)	81 (19.9)	0.012
Fully vaccinated	24 (33.3)	35 (43.2)	0.246
Unvaccinated	48 (66.7)	46 (56.8)	—

COPD, chronic obstructive pulmonary disease. Data are *n* (%) or median (95% interquartile range). ^1^ Delta period: 1 October to 23 November 2021. Omicron period: 15 January to 8 April 2022. ^2^ Due to multiple comparisons, *p*-value < 0.01 was considered significant (marked in bold). ^3^ Patients hospitalized due to COVID-19/Patients hospitalized with SARS-CoV-2 positive PCR. ^4^ See Appendix A for definitions of chronic illnesses. ^5^ These included active cancer treatment or newly diagnosed cancer in the past 6 months (*n* = 36), active hematologic cancer (*n* = 19), previous solid organ transplant (*n* = 17), hematopoietic stem cell transplant within the last 2 years (*n* = 2), active treatment with immunosuppressive medication (*n* = 12), and previous splenectomy (*n* = 3). One patient had both hematologic and solid organ cancer. ^6^ Patient received the second dose of BNT162b2, mRNA-1273, or ChAdOx-1S or first dose of Ad.26.COV2.S at least 14 days before symptom onset. ^7^ This was assessed using the World Health Organization COVID-19 Clinical Progression Scale [21]. ^8^ Some patients received high dose methylprednisolone (1 mg/kg daily) after first receiving dexamethasone (6 mg daily) because of deteriorating respiratory status.

**Table 2 jcm-12-00961-t002:** Association between prior vaccination against SARS-CoV-2 and progression to critically severe disease (the World Health Organization’s COVID-19 Clinical Progression Scale 7–10) [21].

Characteristics	Odds Ratio (95% CI)	*p*-Value ^1^
Intercept	0.08 (0.03–0.20)	**<0.001**
Vaccination status (vaccinated vs. unvaccinated)	0.42 (0.26–0.68)	**<0.001**
SARS-CoV-2 variant (Omicron vs. Delta)	0.77 (0.52–1.14)	0.187
Age	1.02 (1.00–1.03)	**0.038**
Sex (male vs. female)	1.33 (0.97–1.82)	0.074
Charlson comorbidity index	1.11 (1.01–1.22)	**0.035**
Immunocompromising condition present (yes vs. no)	1.32 (0.75–2.33)	0.339
Vaccine status—SARS-CoV-2 variant interaction	1.80 (0.93–3.47)	0.079

CI, confidence interval. ^1^
*p*-value < 0.05 was considered significant (marked in bold).

**Table 3 jcm-12-00961-t003:** Cause-specific hazard ratios for discharge from hospital.

Characteristics	Hazard Ratio (95% CI)	*p*-Value ^1^
SARS-CoV-2 variant (Omicron vs. Delta)	1.13 (0.94–1.36)	0.186
Vaccination status	1.39 (1.13–1.72)	**0.002**
(vaccinated vs. unvaccinated)		
Age	0.99 (0.99–1.00)	0.110
Sex (male vs. female)	0.91 (0.79–1.06)	0.221
Charlson comorbidity index	0.96 (0.91–1.01)	0.115
Immunocompromising condition present (yes vs. no)	0.70 (0.53–0.93)	**0.013**
Vaccine status—SARS-CoV-2 variant interaction	0.92 (0.68–1.25)	0.608

CI, confidence interval. ^1^
*p*-value < 0.05 was considered significant (marked in bold).

**Table 4 jcm-12-00961-t004:** Characteristics of patients hospitalized for vaccine breakthrough COVID-19 by study period.

Characteristic	Delta Period *n* = 175/529 (33.1%)	Omicron Period *n* = 159/407 (39.1%)	*p*-Value ^1^
Age	74 (64–83)	78 (71–84.5)	**0.003**
Men	110 (62.9)	87 (54.7)	0.148
Charlson comorbidity index	4 (3–6)	5 (4–7)	**<0.001**
Immunocompromising condition	37 (21.1)	20 (12.6)	0.042
Received booster dose	1 (0.6)	55 (34.6)	**<0.001**
Critically severe disease (WHO score 7–10) ^2^	33 (18.9)	42 (26.4)	0.115
Death (WHO score 10) ^2^	24 (13.7)	35 (22)	0.061
Time since primary vaccination in weeks	27 (21.3–33.5)	37.9 (27–47.1)	**<0.001**
Time since last vaccination in weeks	27 (21.1–33.4)	23.1 (16.5–35.7)	**<0.001**

^1^*p*-value < 0.05 was considered significant (marked in bold). ^2^ This was assessed using the World Health Organization COVID-19 Clinical Progression Scale [21].

## Data Availability

We would like to thank all healthcare professionals and laboratory personnel involved in managing patients with COVID-19 at the University Medical Center Ljubljana, Slovenia.

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
