# Peer review of "SARS-CoV-2 Vaccination and Clinical Presentation of COVID-19 in Patients Hospitalized during the Delta- and Omicron-Predominant Periods"

_jcm, 2023, doi:10.3390/jcm12030961_

Round 1

Reviewer 1 Report

The article

 SARS-CoV-2 Vaccination and Clinical Presentation of COVID-2 19 in Patients Hospitalized During the Delta and Omicron Pre-3 dominant Periods” compares the effectiveness of monovariant vaccine formulations against Omicron and Delta vatrient of COVID-19. It already cover 407+529 patients for comparison with all demographic information and treatments, which makes it worth reading but I have some minor concerns like:

1.     For Charlson comorbidity, is it fine to compare COPD with other comorbidities in case of COVID-19?

2.     Secondly can we consider Methylprednisolone, dexamethasones and corticosteroids in similar class?

3.     Do you have data for the detail in the pattern of deaths for various treatment therapies and various vaccines or various COVID-variants? This could make it even more interesting.

Author Response

Dear Reviewer 

Please find attached a revised version of our original research article entitled “SARS-CoV-2 Vaccination and Clinical Presentation of COVID-19 in Patients Hospitalized During the Delta and Omicron Predominant Periods”. We thank you for the constructive comments and suggestions, permitting us to improve the manuscript. Our detailed responses are listed below.

Reviewer: The article “SARS-CoV-2 Vaccination and Clinical Presentation of COVID-2 19 in Patients Hospitalized During the Delta and Omicron Predominant Periods” compares the effectiveness of monovariant vaccine formulations against Omicron and Delta vatrient of COVID-19. It already cover 407+529 patients for comparison with all demographic information and treatments, which makes it worth reading but I have some minor concerns like:

Point 1. For Charlson comorbidity, is it fine to compare COPD with other comorbidities in case of COVID-19?

Response: Chronic lung diseases such as COPD are just one of many types of co-morbidities that have been shown to significantly increase the risk of morbidity and mortality in COVID-19. We detailed the types of co-morbidities to give the reader a better sense of the characteristics of patients included in the study. The Charlson comorbidity index represents a score of 19 comorbidities weighted according to severity and includes COPD (Charlson ME, Pompei P, Ales KL, et al. A new method of classifying prognostic comorbidity in longitudinal studies: development and validation. J Chronic Dis 1987; 40: 373-83). This index was used in our study to classify patients’ co-morbid characteristics in a similar manner to Wong et al (https://www.thelancet.com/action/showPdf?pii=S0140-6736%2822%2901586-0). Corrections made. Please see lines 308-311.

Point 2. Secondly can we consider Methylprednisolone, dexamethasones and corticosteroids in similar class?

Response: We agree with the reviewer that methylprednisolone and dexamethasone have pharmacodynamic and pharmacokinetic differences. For COVID-19, the type of steroid may be less important than the dose-equivalent. In our center, methylprednisolone is usually the corticosteroid prescribed in high doses in more severe cases. Corrections made. Please see Table 1.  

Point 3. Do you have data for the detail in the pattern of deaths for various treatment therapies and various vaccines or various COVID-variants? This could make it even more interesting.

Response: Since autopsies were performed only exceptionally, we do not have detailed data for patterns of deaths beyond detecting cardio-respiratory failure in the diseased. Our study was not designed to assess the association between different treatment regimens or different vaccine types and disease outcome. In trying to address potential pathophysiologic mechanisms underlying disease severity we included data on the frequency of hypoxemia on admission and discussed potential pathogenic differences between the Delta and Omicron COVID-variants. Please see lines 328-330.

Reviewer 2 Report

I have a few recommendations for authors: 1.Please justify the use of the Charlson comorbidity index with appropriate references. Is this index taken into account in the treatment of patients with covid-19 in practice of the hospital.

2.There are missing lines in Table 1: after line “Death” lines “Fully vaccinated” and “Unvaccinated” are needed.

Author Response

Dear Reviewer 

Please find attached a revised version of our original research article entitled “SARS-CoV-2 Vaccination and Clinical Presentation of COVID-19 in Patients Hospitalized During the Delta and Omicron Predominant Periods”. We thank you for the constructive comments and suggestions, permitting us to improve the manuscript. Our detailed responses are listed below.

Reviewer: I have a few recommendations for authors: 

Point 1. Please justify the use of the Charlson comorbidity index with appropriate references. Is this index taken into account in the treatment of patients with covid-19 in practice of the hospital.

Response: The Charlson comorbidity index was not used for selecting different treatments for patients with Covid-19 in our hospital. The Charlson comorbidity index represents a score of 19 comorbidities weighted according to severity (Charlson ME, Pompei P, Ales KL, et al. A new method of classifying prognostic comorbidity in longitudinal studies: development and validation. J Chronic Dis 1987; 40: 373-83). This index was used in our study with the intention of providing a more detailed description of the severity of patients’ co-morbid characteristics as used e.g. by Wong et al (https://www.thelancet.com/action/showPdf?pii=S0140-6736%2822%2901586-0). As suggested by the reviewer, we added a reference for it. Please see lines 308-311.

Point 2. There are missing lines in Table 1: after line “Death” lines “Fully vaccinated” and “Unvaccinated” are needed.

Response: Corrections made. Please see Table 1.
